# Experimental Verification and Comparative Analysis of Equivalent Methods on Metal’s Fixed Joint Interface

**DOI:** 10.3390/ma12152381

**Published:** 2019-07-26

**Authors:** Renxiu Han, Guoxi Li, Jingzhong Gong, Meng Zhang, Kai Zhang

**Affiliations:** 1College of Intelligence Science and Technology, National University of Defense Technology, Changsha 410073, China; 2School of Mechanical Engineering, Hunan International Economics University, Changsha 410205, China

**Keywords:** metal joint interface, virtual material method, modal experiment

## Abstract

In order to effectively improve the dynamic characteristics of the fixed metal joint interface, it is important to establish a correct equivalent model of the metal joint interface. In this paper, three equivalent methods for simulating the metal joint interface are analyzed, including the virtual material method, spring damping method, finite element method, and verification by modal experiment. First, according to the contact mechanics model of the constructed metal joint interface, the physical properties of the three-dimensional models of the fixed joint interface are assigned in the ANSYS software. Then, three methods are used for the modal analysis and compared with a modal experiment. The results show that the modal shapes of the three theoretical methods are consistent with those of the experimental modes. The first five natural frequencies obtained by the virtual material method are closest to the experimental natural frequencies, and the errors are within 10%. The errors of the other two methods are between 9% and 39%. Therefore, the virtual material method is a better equivalent method of the metal joint interface.

## 1. Introduction

The metal surface machined by the machine tool is a rough surface composed of many microscopic irregular peaks and valleys. Studies have shown that [1,2,3,4], although the surface morphology produced by a machine tool shows complex and random features in local minute details, there are still certain rules to follow. The machined metal surface has self-affine fractal features that can be described by fractal geometry. Therefore, the metal’s fixed joint interface consists essentially of two rough surfaces with fractal properties. The development of contact mechanics for rough surfaces originated from the ideal smooth surface frictionless elastic contact theory proposed by Hertz in 1881, but contact mechanics developed rapidly within decades. Based on the Hertz contact theory, Majumdar and Bhushan [5] proposed the elastoplastic contact fractal geometry theory of rough surface with scale independence, the MB model, in 1991. The MB model uses the fractal parameters to quantitatively express the contact area and contact load of the metal’s fixed joint interface.

Through the study of the joint surface performance, scholars found that 50% of the system stiffness and more than 90% of the system damping come from the fixed joint interface [6]. Therefore, the selection of the exact equivalent method of the metal joint interface is of great significance for engineering applications. The precise equivalent method of the joint interface can solve the engineering practicality problem of the dynamic model. The current equivalent methods of fixed metal joint interface include the virtual material method (VMM), spring damping method (SDM) and finite element method (FEM). The virtual material method is the most popular equivalent method of the joint interface in recent years.

The VMM, which is also called the equivalent gap method [7], was proposed by Tian [8]. Because the joint surface has the characteristics of gap, friction, wear and hysteresis, the method is proposed from the microscopic point of view, considering the elastoplastic deformation of the real contact surface, the material variation characteristics, thermal characteristics and timeliness of the contact metal. The virtual material is composed of a rough metal surface and a medium of contact surface, which is the main deformation area. The method uses the fractal theory [9] to establish the virtual material model of the joint interface, and to calculate the parameters of the model from the microscopic point of view, such as the Poisson ratio, density and elastic modulus of the virtual material. The length and width of the virtual material are the length and width of the joint, respectively. The thickness of the virtual material is determined according to the actual gap between the two surfaces. The virtual material and the fixed joint are fixedly connected on both sides, and finally applied to the finite element analysis. Zhao [10] improved the VMM, and adopted the improved method to simulate the mechanical joint interface. As a result, the accuracy of the results was improved. Huang [11] applied the virtual material method to simulate the bolt joint interface in the modal analysis. The SDM is used to characterize the stiffness and damping between mechanical joints by providing spring damping elements on the nodes of joint interfaces [12] to achieve the equivalent mechanical joint interface. The selection of the nodes depends on the shape of the fixed joint interface, the connection mode of the fixed joint interface, and the area of the fixed joint interface. The modal analysis of the method in the free state proves the practicability and accuracy of the method.

The accurate analysis of the dynamic characteristics of the joint interface is of great significance for the optimization of the dynamic performance of the whole machine. Therefore, it is necessary to select a high-precision equivalent method of the joint interface. In this paper, the modal analysis of the joint interface is carried out by a modal experiment. The qualitative comparison of the mode shape and the quantitative comparison of the natural frequency are used to verify the feasibility of the joint equivalent method and to select a better equivalent method of the metal joint interface.

## 2. Materials and Methods

Two identical square plates are selected as experimental specimens. The specimen, made of 45#steel, has 200 mm length, 6 mm thickness, 7850 kg/m^3^ density, 3.2 µm surface roughness, 209 Gpa elastic modulus, and 0.3 Poisson ratio. The two plates are pressed by a pressure of 0.625 MPa, as shown in Figure 1. Three methods which are equivalent to the joint interface of the above conditions are as follows:

### 2.1. Virtual Material Method

The VMD divides the three-dimensional model of the two boards into upper, middle, and lower boards (numbers 1, 2, and 3, respectively) as shown in Figure 2. The thickness of the board 1 and board 3 is $h_{1}=h_{2}=5.6\;\mathrm{mm}$. Their physical properties are unchanged. The board 2 is the virtual material. By theoretical calculation, the thickness of the board 2 is h=0.8 mm [8], and the formula for calculating the elastic modulus of the virtual material can be written as [10]:(1)$$E=\frac{2^{D-4.5}E^{*}D\varphi^{1-\frac{D}{2}}G^{1-D}{a_{L}^{\prime }}^{\frac{D}{2}}}{3\pi^{\frac{1+D}{2}}{\left(\ln\;\gamma \right)}^{\frac{1}{2}}A_{0}}\left({a_{L}^{\prime }}^{\frac{1}{2}}-{a_{C}^{\prime }}^{\frac{1}{2}}\right)$$where D and G are the fractal parameters of the surface profile of the joint interface, φ describes the domain extension factor for the micro-contact size distribution associated with D, γ is the scaling parameter, $a_{L}^{\prime }$ is the truncated area of the largest elastic micro-contact, $a_{c}^{\prime }$ is the critical truncated area demarcating the elastic and plastic deformation regimes, $A_{0}$ is the nominal contact area and $E^{*}$ is the equivalent elastic modulus. According to [8], D and G are determined by the roughness of the surface of the joint interface. According to [10], $a_{L}^{\prime }$ is only affected by the surface pressure exerted on the board when the physical properties of the material are unchanged. The formula of the equivalent elastic modulus can be written as [13]:(2)$$\frac{1}{E^{*}}=\frac{\left(1-\nu_{1}^{2}\right)}{E_{1}}+\frac{\left(1-\nu_{2}^{2}\right)}{E_{2}}$$where $\nu_{1}$ and $E_{1}$ are the Poisson’s ratio and elastic modulus of the board 1 and board 2, and $\nu_{2}$ and $E_{2}$ are the Poisson’s ratio and elastic modulus of the board 3. 

The normal stiffness value between the joint interfaces and the shear modulus of the virtual material can be described as [14](3)$$Kn=EA_{0}/h$$
(4)$$G_{\tau }=K_{\tau }h/A_{0}$$where $K_{\tau }$ is the tangential contact stiffness. According to [15], the ratio of the tangential contact stiffness to the normal contact stiffness is 0.25 to 0.35. Without a loss of generality, $K_{\tau }/K_{n}$ is taken as 0.35. 

The Poisson’s ratio and density of the virtual material can be described, respectively, as [10,16]:(5)$$\upsilon =\frac{E}{2G_{\tau }}-1$$
(6)$$\rho =\frac{\rho_{1}h_{1}+\rho_{2}h_{2}}{h_{1}+h_{2}}$$where $\rho_{1}$ and $\rho_{2}$ are the density of board 1 and board 2, respectively. Through the above calculation, E=0.3554 Gpa, ν=0.429 and $\rho =7850\;\mathrm{kg}/m^{3}$.

### 2.2. Spring Damping Method and Finite Element Method

In the SDM model, a set of twelve spring damping units connects to the four corners of the joint interface, so that the joint interface is evenly stressed. The schematic diagram of the model is shown in Figure 3a. At each node, there are three sets of spring damping units. The normal and tangential total stiffness and damping of the spring are the normal and tangential total stiffness and damping of the joint, respectively. The value of each spring and damping can be expressed, respectively, as:(7)$$K_{ix}=K_{iy}=K_{\tau }/4$$
(8)$$K_{iz}=K_{n}/4$$
(9)$$C_{ix}=C_{iy}=C_{\tau }/4$$
(10)$$C_{iz}=C_{n}/4$$where $K_{\tau }$ is the tangential contact stiffness, $K_{n}$ is the normal stiffness, $C_{\tau }$ is the tangential damping and $C_{n}$ is the normal damping. According to [17], the stiffness value and damping value of the spring damping unit can be obtained: $K_{x}=K_{y}=8.53e^{7}\;N/m$, $K_{z}=2.44e^{8}\;N/m$, $C_{x}=C_{y}=5.22e^{5}\;N\cdot s/m$ and $C_{z}=4.62e^{8}\;N\cdot s/m$.

The FEM connects the three-dimensional model of the joint interface through the bond connection method in the ANSYS software, as shown in Figure 3b, so that the two boards become a whole. The original physical properties of the board were used, and the finite element analysis was performed. Therefore, the elastic modulus $E_{0}$, the Poisson ratio $\nu_{0}$ and the density $\rho_{0}$ of the board of the FEM are as follows: $E_{0}=209\;\mathrm{Gpa}$, $\nu_{0}=0.3$ and $\rho_{0}=7850\;\mathrm{kg}/m^{3}$.

## 3. Simulation, Modal Experiment and Results

### 3.1. Modeling and Simulation

The above three models are modeled in three dimensions through the CATIA software. The model was imported into the ANSYS or ABAQUS software, the boundary condition was set to the free state, and the physical properties of the model were assigned. The mesh diagram of the model of the FEM is shown in Figure 4. The grid has a total of 129,955 nodes and 63,599 elements. The six holes in the Figure 4 are bolt holes. A force was applied around the bolt holes so that the pressure on the joint interface is equal to 0.625 MPa. Finally, the modal analysis is performed. The first five natural frequencies are shown in Table 1, and the mode shapes are shown in Table 2.

### 3.2. Modal Experiment

The modal experiment (ME) uses an exciter to excite and laser scans to acquire data. The natural frequencies and mode shapes of the test piece are calculated and analyzed based on the measured response data. By comparing the results obtained by finite element analysis and modal experiments, the optimal equivalent method of the joint interface can be obtained.

First, a torque wrench was used to fasten two identical boards together by bolts, as shown in Figure 5a. Surface pressure is applied to the plates through the bolts. In order to measure the natural frequencies of the joint interface, a boundary-less condition is selected. However, it cannot be fully realized in the experiment. Therefore, the experiment chooses two elastic ropes to hang the two boards in front of the exciter. The specimens are vibrated by the exciter. The laser is used to detect the modal shapes and natural frequencies of the specimen. The final data is collected on the computer for signal processing and identification. The natural frequencies and mode shapes of the first five orders are obtained. The experimental setup is shown in Figure 5b.

### 3.3. Results

The first five natural frequencies of the modal experiment are 660.9 Hz, 1132.8 Hz, 1325.0 Hz, 1564.1 Hz, 1564.1 Hz, 1635.1 Hz, respectively, as shown in Figure 6. The first five modal shapes obtained by the experiment are shown in Table 2.

## 4. Discussion

The comparison between the first five-order mode shapes of the modal experiment and those of the simulation is shown in Table 2. Compared with the experimental mode shape diagrams, the mode shapes of the three equivalent methods of the joint interface are roughly similar to the experimental mode shapes. However, the second-order mode shape obtained by the SDM is significantly different from the mode shape obtained by the experiment. Furthermore, the mode shapes obtained by simulation analysis are slightly different from those obtained by the experiment in the edge area of the mode shape, especially the fifth mode shape. The reason for this is that the test area is the red border area, as shown in Figure 5a, and the mode shapes outside the red frame cannot be measured. Therefore, it can be proved that the VMD and the FEM are more suitable for simulating the metal’s fixed joint interface.

The contrast and error of the first five natural frequencies obtained by simulation and modal experiment are shown in Table 3 and Figure 7. By comparing the first five natural frequencies obtained by the modal experiment, it is found that the error of the natural frequencies obtained by the VMM is the smallest, being within 10%. The errors of the natural frequencies obtained by the other two equivalent methods of the joint interface are between 9% and 39%. The results prove that the VMM is more realistic than the other two methods. It can be clearly seen from Figure 7 that, as the order increases, the difference between the natural frequency obtained by the simulation and the modal experiment is larger. Therefore, it can be explained that for a high-order modal simulation, the equivalent methods for simulating the metal joint interface will lose their authenticity.

The three equivalent methods of the joint interface all have disadvantages. The VMD and the SDM are the joint interface equivalent models established based on the fractal contact theory. By modelling the contact mechanics of the joint interface, the physical properties of the virtual material, and the stiffness and damping values of the spring damping unit, are obtained respectively. However, the fractal contact theory has certain limitations. The contact mechanics model ignores the interaction between the asperities. When the squeezing force is small, this assumption is approximately true. However, when the squeezing force is relatively large, the interaction between asperities cannot be ignored. Therefore, it will affect the establishment of the equivalent method of the fixed joint interface. The VMM is also affected by the thickness of the virtual material. If the thickness of the virtual material is too large or too small, the VMM will lose some authenticity. The SDM cannot accurately determine the number and position of the spring damping unit. Generally, the number and position of the spring damping unit are determined according to the number of nodes of the joint interface, but it also causes a certain error. For example, the second-order mode shape of the SDM does not match the experimental mode shape. The FEM ignores the tiny structure between the joint interface so that the modal simulation brings a big error. Additionally, the simulation environment cannot be exactly the same as the experiment. Therefore, there are certain errors in the equivalent methods of the joint interface.

By analyzing the error of the results obtained by the modal experiment and the modal simulation, the following aspects can be improved. First, in the basic theory part, it is necessary to consider the interaction between the asperities of the fixed joint interface to establish a more precise contact mechanics model. Second, the multi-variable modal simulation is carried out on the thickness of the virtual material and the number and position of the spring damping unit, and then compared with the modal experiment to obtain a more accurate equivalent method on the metal’s fixed joint interface.

## 5. Conclusions

In this paper, the modal analysis of the VMM, the FEM and the SDM was carried out by the finite element simulation software. The first five natural frequencies and mode shapes of the three joint interface equivalent models were obtained, and then the modal experiment was performed. Finally, the experimental results were compared with the results of the simulation. The conclusions can be summarized as follows:The modal shapes of the modal experiment are roughly similar to the modal shapes obtained by the three equivalent methods of the joint interface. It is proved that all three methods can simulate the microscopic metal joint interface. The natural frequencies obtained by three equivalent methods of the metal joint interface have obvious deviations from the results of the modal experiment. The error of the natural frequency value obtained by the VMM is the smallest, being within 10%. Obviously, it is lower than the errors of the other two methods, which proves that the VMM is the closest to the real situation among the three equivalent methods of the metal joint interface.

## Figures and Tables

**Figure 1 materials-12-02381-f001:**
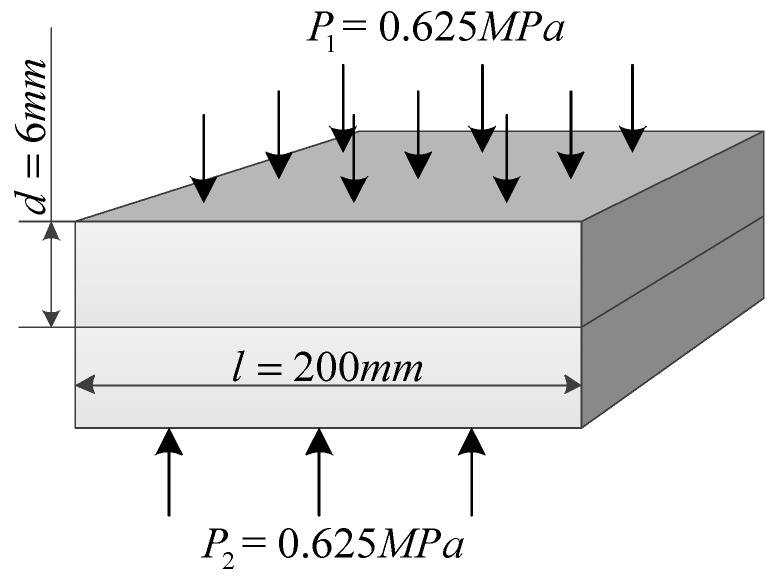
Dimensions of the specimen.

**Figure 2 materials-12-02381-f002:**
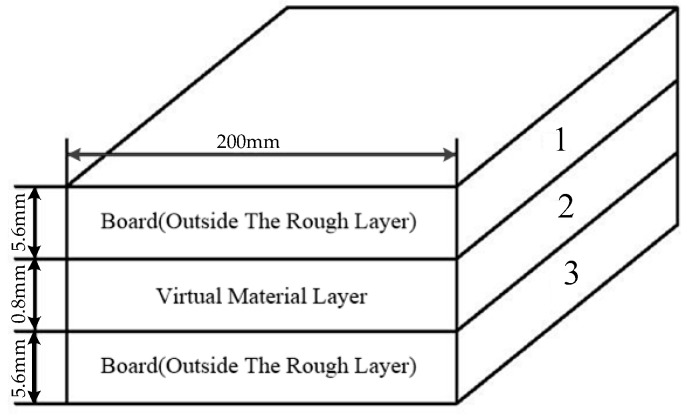
Schematic diagram of the three-dimensional model of VMM.

**Figure 3 materials-12-02381-f003:**
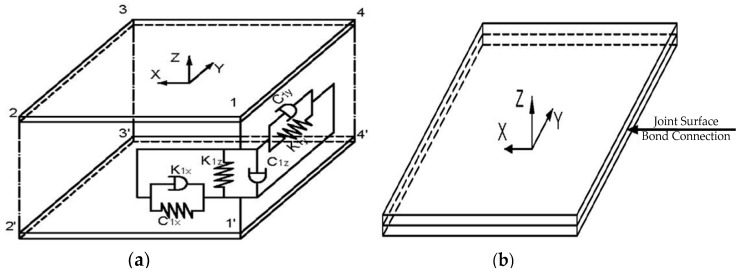
Schematic diagram of three-dimensional model of (**a**) theSDM and (**b**) the FEM.

**Figure 4 materials-12-02381-f004:**
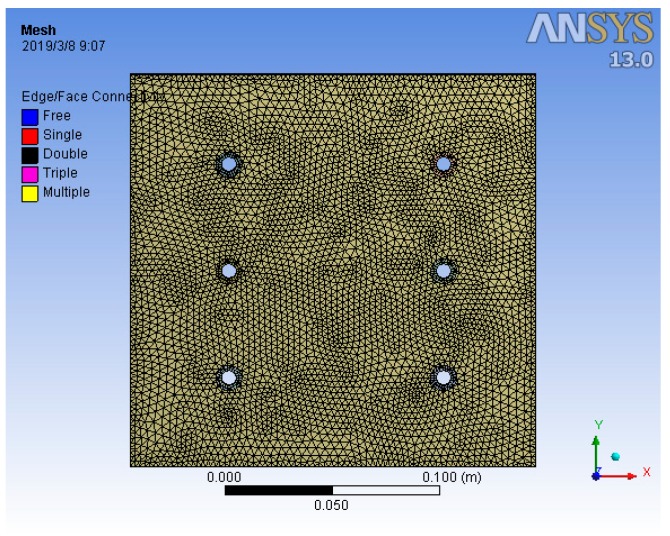
Grid diagram of the three-dimensional model of the FEM.

**Figure 5 materials-12-02381-f005:**
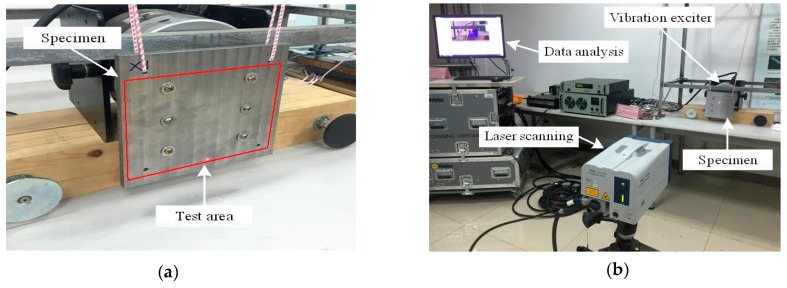
(**a**) Experimental fasten boards, and (**b**) experimental setup.

**Figure 6 materials-12-02381-f006:**
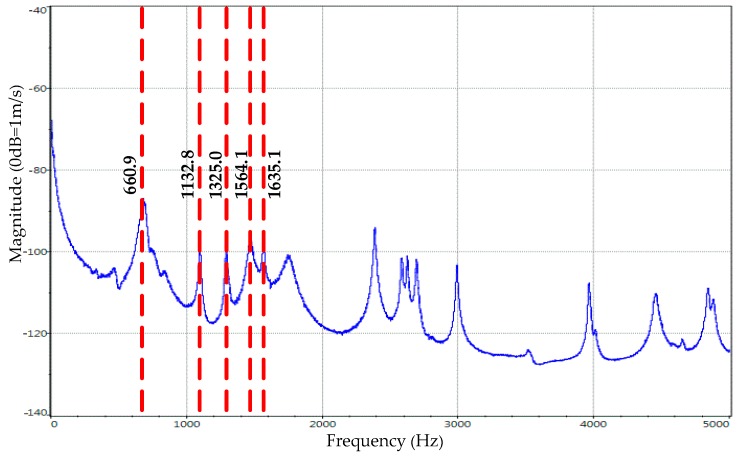
Curve of frequency response of specimen by experiment.

**Figure 7 materials-12-02381-f007:**
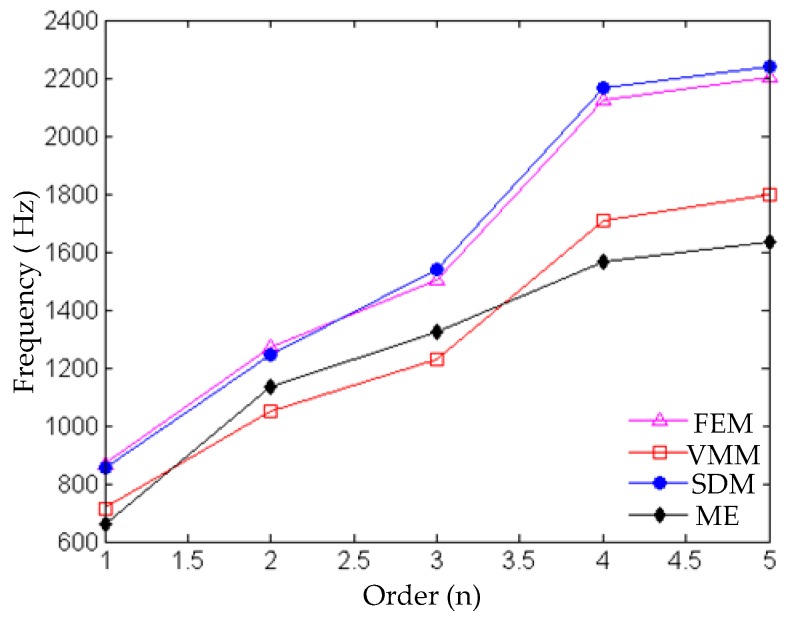
Comparison of the first five frequencies obtained by simulation and experiment.

**Table 1 materials-12-02381-t001:** The first five natural frequencies of the three equivalent methods of the joint interface.

Methods	*f*_1_ (Hz)	*f*_2_ (Hz)	*f*_3_ (Hz)	*f*_4_ (Hz)	*f*_5_ (Hz)
VMM	7140	1049.8	1228.8	1707.1	1790.1
SDM	855.8	1243.3	1540.6	2166.4	2237.8
FEM	867	1270.3	1504.2	2144.1	2211.7

**Table 2 materials-12-02381-t002:** Comparison of the first five mode shapes obtained by the simulation and modal experiment.

Order	ME	VMM	FEM	SDM
1	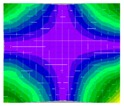	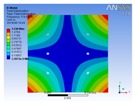	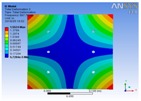	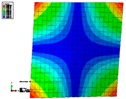
2	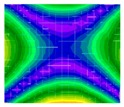	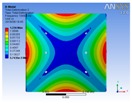	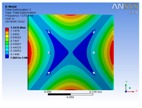	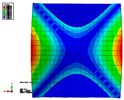
3	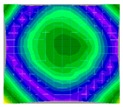	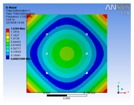	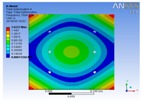	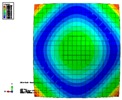
4	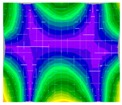	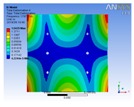	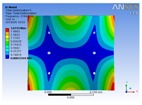	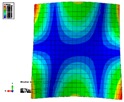
5	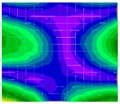	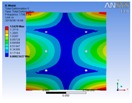	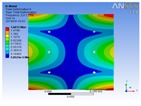	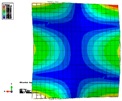

**Table 3 materials-12-02381-t003:** Errors of the first five natural frequencies obtained by simulation and experiment.

Order	1	2	3	4	5
Error of VMM	8.12%	7.32%	7.26%	9.14%	9.93%
Error of FEM	31.18%	12.14%	13.52%	37.08%	35.26%
Error of SDM	29.49%	9.75%	16.27%	38.51%	36.86%

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
