# Peer review of "Experimental Verification and Comparative Analysis of Equivalent Methods on Metal’s Fixed Joint Interface"

_materials, 2019, doi:10.3390/ma12152381_

Round 1

Reviewer 1 Report

Experimental verification and comparative analysis of equivalent methods on fixed joint interface of metal

by Han et al.

The authors present the application of three different models to simulate the experimental behavior of a fixed metal joint interface. The experimental setup is appropriate, and the comparison of experimental results may be of some relevance. Due to these points, the manuscript could deserve publication.

However, some points have to be addressed before publication.

Mayor points:

The Discussion section must be improved trying to discuss deeply the results. For example, nothing is said about the natural frequencies obtained.

Minor points

The manuscript suffers of a poor english level. Please made the following changes:

Title - Change in "Experimental verification and comparative analysis of equivalent methods on metal" fixed joint interface"

Line 36 – Change “The accuracy of the results has improved.” in “As a result, the accuracy of the results was improved.”

Lines 48-51 – Change in “The specimen, made of 45#steel, has 200 mm length, 6 mm thickness, 7850 kg/m3 density, 3.2 mm surface roughness, 209 GPa elastic modulus, and 0.3 Poisson’s ratio.

Line 52 – Change “is” in “are”

Line 62 – In the formula, the prime symbol is in wrong places. Please fix.

Line 67 – Change “According to the reference [2]” in “According to ref. [2]”

Line 68 – Change “According to reference4” in “According to ref. [4]”

Line 75 – Remove “respectively”

Line 78 – Change “According to the reference [9]” in “According to ref. [9]”

Line 89 – Change “The SDM is to connect the twelve sets of” in “ In the SDM model, a set of twelve”

Line 102 – Change “The FEM is to connect” in “The FEM connects” and change “Bond” in “bond”

Line 104 – Change “Set the ….. analysis” in “The original physical properties of the board were used, and the finite element analysis was performed”

Lines 111-113 – Change “Import the…mesh the model” in “The model was imported into ANSYS or ABAQUS software, the boundary condition was setted to free state, and the physical properties of the model were assigned”

Line 114 – Change “Nodes” in “nodes”

Lines 115,116 – Change “Apply a force around the bolt hole” in “A force was applied around the bolt holes”

Line 126 – Change “Firstly, use a torque wrench” in “Firstly, a torque wrench was used”

Line 131 – Change “”is” in “are

Line 133 – Change “device” in “setup”

Line 146 – Change “is” in “are”

Line 147 – Change “shape especially” in “shape, especially”

Conclusions section: after line 168, it is better to mention the different conclusions in a bulleted list.

Reviewer 2 Report

This manuscript the authors compare three different methods to simulate metal joint interfaces. I find the manuscript acceptable for publication. However the authors should modify a bit the introduction to render the manuscript more attractive for a broader audience, since at the moment it is too specialized. For example, the terms metal joint interface and virtual material method are very specialized, so I think that the authors should include in the introduction some basic statements to introduce their topic. I feel that the way that the manuscript is written at the moment will be of interest for a very narrow readership.

Round 2

Reviewer 1 Report

Dear Autors

The manuscript is now suitable for publication.